# Caregivers’ Perceived Impact of WIC’s Temporary Cash-Value Benefit (CVB) Increases on Fruit and Vegetable Purchasing, Consumption, and Access in Massachusetts

**DOI:** 10.3390/nu14234947

**Published:** 2022-11-22

**Authors:** Cristina Gago, Rachel Colchamiro, Kelley May, Eric B. Rimm, Erica L. Kenney

**Affiliations:** 1Department of Nutrition, Harvard T.H. Chan School of Public Health, Boston, MA 02115, USA; 2Massachusetts Department of Public Health, Boston, MA 02108, USA; 3Department of Epidemiology, Harvard T.H. Chan School of Public Health, Boston, MA 02115, USA; 4Channing Division of Network Medicine, Brigham and Women’s Hospital, Harvard Medical School, Boston, MA 02115, USA; 5Department of Social and Behavioral Sciences, Harvard T.H. Chan School of Public Health, Boston, MA 02115, USA

**Keywords:** Special Supplemental Nutrition Program for Women, Infants and Children (WIC), Cash-Value Benefit (CVB), fruit and vegetable consumption, fruit and vegetable affordability, nutrition policy, early childhood, access to healthy foods, healthy nutrition, food assistance, low-income population

## Abstract

Responding to the COVID-19 pandemic, the American Rescue Plan (2021) allowed state agencies of the Special Supplemental Nutrition Program for Women, Infants, and Children (WIC) the option of temporarily increasing the Cash-Value Benefit (CVB) for fruit and vegetable (FV) purchases. To examine the impact of this enhancement on WIC caregiver experience, the MA WIC State Office invited 4600 randomly selected MA WIC caregivers to complete an online survey (February–March 2022). Eligible adults had at least one child, had been enrolled at least a year, and were aware of the increase. Of those who opened the screener (*n* = 545), 58.9% completed it (*n* = 321). We calculated the frequencies of reporting increased FV outcomes and tested whether responses differed by race/ethnicity, market access, and food security. Most caregivers perceived the CVB increase to benefit FV purchasing (amount and quality, 71.0% and 55.5%), FV consumption (offered to children and personally consumed, 70.1% and 63.2%), and satisfaction with the WIC food package (37.1% reported improved satisfaction, pre- vs. post-increase). Probability of reporting improved outcomes was not found to differ by race/ethnicity, market access, or food security. CVB increases may pose important implications for dietary behaviors and satisfaction with WIC. Policymakers should consider making this increase permanent.

## 1. Introduction

Fruit and vegetable (FV) consumption is important for health promotion and chronic disease prevention across the lifespan [1]. However, a majority of U.S. children and adults do not meet recommendations for FV consumption [2], as defined by the Dietary Guidelines for Americans [3]. This is especially true among those residing in low-income communities, where FV access tends to be limited [4,5,6] and food insecurity pervasive [7].

To address poor nutritional access among low-income families with young children, the Special Supplemental Nutrition Program for Women, Infants, and Children (WIC) was first piloted in 1972. Beyond nutrition education, breastfeeding support, and referral services, WIC provides over six million enrolled participants [8] with a monthly “food package” which is disbursed as an Electronic Benefit Transfer (EBT) Card for the purchase of eligible foods from WIC-approved retailers [9]. Since 2009, a Cash-Value Benefit (CVB) [10] has been offered as part of most food packages, which provides participants with a small, fixed dollar allowance to be spent on fruits and vegetables. The CVB has been shown to be a highly valued aspect of the WIC food package by participants [10] and, in combination with other changes to the WIC food packages in 2009, may have played a role in improving WIC participants’ diets [11,12,13].

During the COVID-19 pandemic, due to a provision of the American Rescue Plan Act of 2021 (which established several programs to address food insecurity and other economic consequences of the pandemic), WIC state agencies were provided with the option of temporarily increasing the monetary value of the CVB for fruit and vegetable purchases. In Massachusetts (MA), this amounted to an initial tripling of the CVB from $11 to $35 in June and a second change from $35 to $24–$47 (depending on the package and participant category) in October. This temporary increase was made in an effort to bolster WIC enrollment, retention, and redemption [14], and as a means of addressing widening disparities in nutritional access and food security [15], amidst rising rates of unemployment and inflation [16,17,18,19]. Though nearly 11 million women, infants, and children are eligible for WIC, only about one-in-two of those eligible actually participate in the program, underscoring a critical gap in coverage [8]. Among known barriers to optimal retention and redemption (e.g., dissatisfaction or cultural mismatch with the food package [20,21], or lack of childcare, time, or transit to make WIC appointments [22,23,24]), low CVB value was commonly cited pre-pandemic as a reason for early drop off from WIC while still eligible [10].

Recent studies in California [25,26], North Carolina [27], and Delaware [28] suggest that this temporary increase is highly valued by WIC caregivers and may be linked with improvements in FV affordability, purchasing, and consumption, as well as overall WIC satisfaction. However, research on the perceived impact of this emergent temporary increase remains limited to mostly qualitative studies [25,27,28]. It is also unclear whether any potential benefits or impacts of the CVB increase may have been mitigated by structural inequities in access to FV. To address the diversity of WIC caregiver experience across states and food retail environments, studies from diverse contexts are essential for the evaluation of this temporary benefit enhancement.

To address this need, we aim to evaluate MA WIC caregiver perceptions regarding FV accessibility post-CVB-increase, as well as perceived impact of the CVB-increase on FV behaviors and satisfaction with the WIC food package. Finally, we aim to assess potential disparities in perceived impact by race/ethnicity, market access, and food security status.

## 2. Materials and Methods

### 2.1. Study Design, Setting, and Sample

In February-March 2022, the MA WIC State Office invited 4600 randomly selected MA WIC caregivers via text message to complete an online survey, which took approximately 10 min to complete. Eligible adults spoke English, Spanish, or Portuguese; had at least one child (age 2–5 years) enrolled in WIC for at least a year; and remembered the CVB increase. Of those selected caregivers who opened an eligibility screener survey (*n* = 545), 321 were deemed eligible, provided consent, and completed the full survey (Figure 1). The Harvard T.H. Chan School of Public Health Institutional Review Board approved all study procedures.

### 2.2. Measures

#### 2.2.1. Sociodemographic Characteristics

Caregivers self-reported primary language, age in years and the region of MA in which they lived, as well as caregiver gender, race/ethnicity, marital status, educational attainment, employment status, and food security status (via Hunger Vital Sign, Boston, MA, USA, [29]).

#### 2.2.2. Access to FV

To measure aspects of the accessibility and affordability of caregivers’ food retail environments that might impact CVB utilization, we adapted items previously published by Rose and Richards (2004) [30]. Caregivers were asked to report on the type of store where they do most of their food shopping, how they traveled to the store where they did most of their food shopping (e.g., car, walking), the average length of time needed for this travel (in minutes), and their perceptions of the acceptability and affordability of FV sold at their preferred store.

#### 2.2.3. Primary Outcomes: Perceived Impact of CVB-Increase on FV Behavior and Satisfaction with WIC Food Package

Caregivers also reported whether they perceived an increase, no change, or a decrease in their household’s FV expenses and amount purchased, consumed, and offered to children from before to after the CVB-increase. Lastly, caregivers ranked their satisfaction with FV quality purchased using the enhanced CVB and with the WIC food package overall on a five-point scale, from very satisfied to very unsatisfied, for both before and after the CVB increase. Change in satisfaction with (a) FV quality and (b) the WIC food package overall was coded by comparing caregivers’ reported recalled satisfaction before the CVB increase to their reported satisfaction after the increase, and categorized as increased satisfaction, no change, or decreased satisfaction.

### 2.3. Statistical Analysis

We calculated frequencies of responses for our primary and secondary outcomes overall, and stratified by race/ethnicity (Hispanic, non-Hispanic (NH) Black, NH White, Other), market travel time (<15 min, ≥15 min), and food security status (food secure, food insecure). We also used χ^2^ tests to evaluate between-group differences. We calculated Poisson regression models (adjusting for age and education) with robust error variance to estimate the relative risk associated with NH Black or Hispanic race/ethnicity compared to NH White race of caregivers perceiving improvements in FV behavior and satisfaction from before to after the CVB increase (compared with perceiving no change or worsening). We calculated similar relative risks assessing associations with travel time to nearest FV market (as a measure of overall access issues) and food security status (based on responses to the two-item Hunger Vital Sign [29] measure). Alpha was set at 0.05; analyses were completed in SAS (v9.4, SAS Institute Inc., Cary, NC, USA). Measures, data, and code are publicly available on Open Science Framework.

## 3. Results

Most caregivers spoke English (68.5%), and most identified as women (96.3%, Table 1). Approximately half were under age 35 (54.2%). In terms of the racial/ethnic distribution in the sample, caregivers most frequently reported identifying as Hispanic (46.4%), followed by NH White (27.4%) and NH Black (12.1). About two-in-five attained at most a high school degree (39.5%) and reported being unemployed (41.1%).

A majority reported they could buy enough FV at their usual market in the month preceding survey completion (61.7%; Table 2). Approximately half lived less than 15 min from their preferred market (53.9%), which most identified as a supermarket (68.9%) reached by car (69.5%). One-fourth reported FV prices to be expensive (28.0%); about half estimated spending more than $100 out-of-pocket (i.e., not using their WIC or SNAP benefits) in the past month on FV (46.1%).

When comparing their experiences pre- vs. post-CVB-increase, most reported decreased out-of-pocket FV costs (70.4%), as well as an increased amount of FV purchased (71.0%), personally consumed (63.2%) and offered to their children (70.1%, Table 3). When reflecting on changes in the types of FV purchased (pre- vs. post-CVB-increase), most caregivers reported purchasing increased amounts of fresh fruits and/or vegetables (72.6%), though the same or reduced amounts of canned (68.8%) and frozen FV (65.7%), indicating the increase in overall FV purchasing was driven by fresh FV purchasing. Over half reported increased satisfaction with purchased FV taste and quality (55.5%). While a majority (65.1%) reported they were already satisfied with the WIC food package pre-CVB-increase, more than one-in-three (37.1%) reported perceived increases in satisfaction following the benefit enhancement.

The probabilities of reporting perceived improvements in FV behavior and satisfaction were not found to differ significantly by race/ethnicity (NH Black vs. NH White, Hispanic vs. NH White) or market access (i.e., <15 min vs. ≥15 min market travel), with a couple exceptions. Hispanic caregivers were less likely to report decreased out-of-pocket costs (RR = 0.86 (95%CI = 0.74, 1.00)) and less likely to report increased satisfaction with the WIC food package, compared with NH White caregivers (RR = 0.66 (95%CI = 0.47, 0.94)); this may have been due to higher baseline levels of satisfaction with the WIC food package pre-CVB-increase for Hispanic caregivers, leaving less room for improvement (38.9% Hispanic reported being very satisfied with the WIC food package pre-CVB, vs. 33.3% NH Black and 27.3% NH White respondents).

## 4. Discussion

This study is among the first to quantitatively examine WIC caregiver experiences with the temporary CVB enhancement initially funded through the American Rescue Plan Act of 2021. Though caregivers reported already high satisfaction with the WIC food package pre-CVB-increase, one-in-three still reported increased satisfaction post-CVB-increase. Most caregivers reported perceived reductions in out-of-pocket costs and increased FV amount purchased, personally consumed, and offered to children; this aligns with findings from other states [27]. With WIC serving over 6 million participants annually [8], the CVB benefit increase thus may have resulted in largescale improvements in FV consumption; this is especially important with most adults and children in the U.S. not meeting recommendations for FV consumption [2]. Given this public health nutrition challenge, CVB may be a promising population-level policy to target widespread FV purchasing and consumption.

Moreover, this study found that the perceived benefits of the CVB increase on FV purchasing and FV consumption appear to have been equitably distributed in terms of race/ethnicity, caregivers’ food access environments, and food security status, as others have reported [26]. The differences by race/ethnicity in perceived benefit to reduced out-of-pocket cost may be reflective of the differences in average monthly grocery expenses reported by the families in this sample; Hispanic caregivers were more likely to report spending over $201/month out-of-pocket on groceries (post-CVB-increase) than NH White or NH Black caregivers (18.1%, 12.8%, 6.8%, respectively), and therefore may have been less sensitive to the financial benefit of the CVB enhancement. Second, although we observed differences in probability of caregivers reporting increased satisfaction post-increase by race/ethnicity (Hispanic vs. NH White), this is likely the consequence of a ceiling effect introduced by the higher levels of satisfaction pre-CVB-increase among Hispanic (vs. NH White) caregivers.

Given the importance of promoting equity in nutrition and healthy eating [31], the finding that most outcomes did not differ by market access, food security status, nor caregiver race/ethnicity is encouraging. With innovations in nutrition or obesity-related policies and programs, there is often a concern that any benefits from a new policy or program may not be equitably accessed by communities of color or individuals with lower incomes [31], and thus may inadvertently exacerbate health disparities by channeling more benefits to those who are already advantaged. Our findings suggest that the CVB increase did not do this.

These findings align with those recently reported from studies conducted in other states, with unique retail contexts and diverse participant needs [25,26,27,28]. Studies to date on caregiver perceptions of this temporary increase demonstrate a consensus: WIC caregivers associate this increase in CVB with an increase in FV purchasing and consumption, as well as an increase in satisfaction with the WIC food package [25,26,27,28]. CVB may thus serve not only as a nutrition policy tool for increasing FV consumption among households at high risk of low FV consumption due to cost and access, but CVB may also as a strategic target for policy and advocacy efforts in addressing suboptimal WIC retention and redemption [32]. Though participation in WIC is associated with improved diet and health in early childhood [33], suboptimal retention and redemption [32] prevents WIC from reaching its full potential. This research suggests that making the CVB increase permanent serve as a strategy to help address two important public health nutrition goals: (1) increasing FV consumption and (2) helping to promote satisfaction, and subsequent retention, in WIC. Future research should more thoroughly evaluate the role that CVB increases may play in influencing WIC retention and redemption of benefits more directly.

Strengths of this study include the participation of a random sample of diverse MA-WIC-enrolled caregivers, as well as the fact that we were able to roll out the survey soon after the CVB increase had gone into effect, so that participants would have better recall about the change. As a preliminary study of this policy change, in which we quickly deployed a low-burden survey to capture participants’ perceptions, our study also has several limitations. First, because the survey assessed caregivers’ perceptions of the program, we cannot say for sure that the improvements seen in this study translate to actually higher FV benefit redemption and consumption. Future research should leverage redemption data and/or dietary intake data to answer these questions more fully. The cross-sectional design also relies on participant recollection (regarding WIC experiences day of survey vs. year prior), as data pre-CVB increase was not collected. Our response rate was also lower than anticipated, which may have been due to our use of text-based recruitment and online survey tools [34]; however, these methods were selected to help maintain participant privacy, reduce risk of social desirability bias, and increase reach (e.g., offering flexible response languages and times). In spite of these challenges, we were still able to recruit a representative sample, compared to MA and national WIC populations [35]. However, the small subsample sizes following stratification by race/ethnicity, market travel time, and food security status limited our power to detect differences by groups; these comparisons should be repeated in future studies with larger samples.

## 5. Conclusions

A majority of caregiver respondents enrolled in MA WIC reported that, as a result of the CVB benefit increase during the COVID-19 pandemic, they perceived benefits to FV affordability, amount purchased, and amount consumed (personally and offered to children). In addition to these positive changes in perceived consumption and purchasing behaviors for FV, caregivers also reported perceived increases in satisfaction with the WIC food package. Few differences in the probability of reporting improved outcomes were observed by race/ethnicity, market access, or food security status, though future research is necessary to confirm. These preliminary findings add to a growing body of literature in support of making this temporary benefit enhancement permanent [25,26,27,28], as a means of ensuring that the WIC food package continues to meet the evolving needs of low-income women, infants, and children in the United States.

## Figures and Tables

**Figure 1 nutrients-14-04947-f001:**
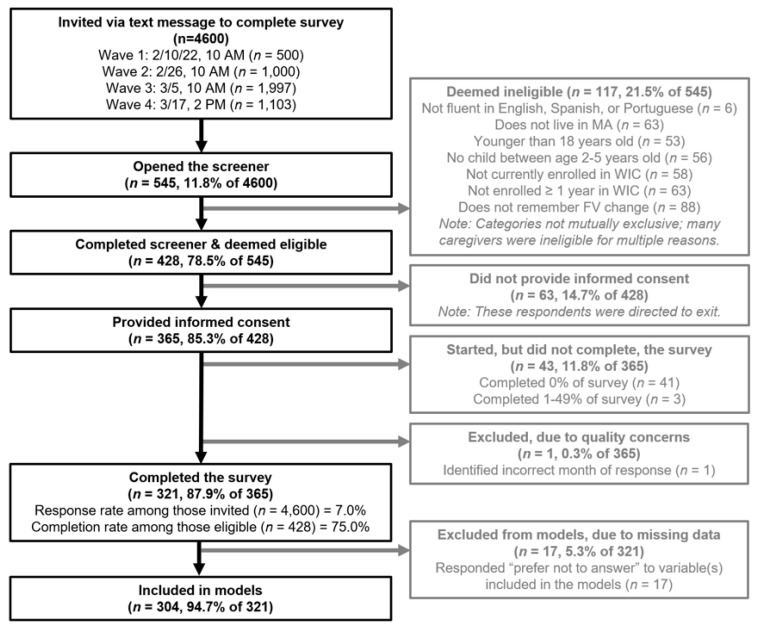
Participant Flow Diagram.

**Table 1 nutrients-14-04947-t001:** Sample Characteristics of Survey Participants, by Race/Ethnicity, Average Market Travel Time, and Food Security Status (*n* = 321 MA WIC caregivers, 2022).

Sociodemographic Characteristics	All	By Race/Ethnicity					By Market Travel Time			By Food Security Status ^e^		
		Hispanic	NH Black	NH White	Other ^d^	*p* ^f^	<15 min	≥15 min	*p* ^f^	Food Secure	Food Insecure	*p* ^f^
	(*n* = 321)	(*n* = 149)	(*n* = 39)	(*n* = 88)	(*n* = 36)		(*n* = 173)	(*n* = 148)		(*n* = 153)	(*n* = 168)	
**Language ^a^**						**<0.001**			**0.05**			**<0.001**
English	68.5	38.9	92.3	96.6	91.7		72.3	64.2		78.4	59.5	
Spanish	24.3	51.0	.	.	2.8		19.1	30.4		13.1	34.5	
Portuguese	7.2	10.1	7.7	3.4	5.6		8.7	5.4		8.5	6.0	
**Residence**						**0.04**			0.16			0.45
Eastern MA	51.1	44.3	61.5	60.2	44.4		56.1	45.3		49.0	53.0	
Central MA	29.6	34.2	33.3	20.5	27.8		26.6	33.1		28.8	30.4	
Western MA	19.3	21.5	5.1	19.3	27.8		17.3	21.6		22.2	16.7	
**Age (in years)**						0.32			0.61			0.01
18–24	5.0	7.4	5.1	3.4	.		6.4	3.4		6.5	3.6	
25–29	21.5	23.5	15.4	22.7	22.2		23.1	19.6		16.3	26.2	
30–34	27.7	27.5	41.0	20.5	22.2		25.4	30.4		23.5	31.6	
35–39	23.7	21.5	23.1	25.0	33.3		23.7	23.7		24.8	22.6	
≥40	22.1	20.1	15.4	28.4	22.2		21.4	23.0		28.8	16.1	
**Gender**						0.28			> 0.9			> 0.9
Woman	96.3	98.0	97.4	96.6	94.4		97.1	95.3		96.7	95.8	
Man	2.5	2.0	2.6	1.1	5.6		2.3	2.7		2.6	2.4	
Other	0.6	.	.	2.3	.		0.6	0.7		0.7	0.6	
Prefer not say	0.6	.	.	.	.		.					0.86
**Marital status ^b^**						0.17			0.68			
Married	44.6	43.6	30.8	54.6	47.2		45.7	43.2		45.1	44.1	
Single	44.2	44.3	56.4	39.8	47.2		43.4	45.3		45.8	42.9	
Prefer not say	11.2	12.1	12.8	5.7	5.6		11.0	11.5		9.2	13.1	
**Educational attainment**						**<0.001**			**0.004**			**0.04**
<High school degree	8.7	16.8	5.1	1.1	.		4.1	14.2		8.5	8.9	
High school degree or equivalent	30.8	31.5	33.3	25.0	44.4		28.3	33.8		28.1	33.3	
Some college (no degree)	29.9	26.9	25.6	39.8	25.0		31.2	28.4		26.8	32.7	
≥College degree	25.6	20.8	30.8	33.0	25.0		30.1	20.3		33.3	18.5	
Prefer not say	5.0	4.0	5.1	1.1	5.6		6.4	3.4		3.3	6.6	
**Employment status ^c^**						0.9			0.27			0.49
Employed part time	31.2	31.5	30.8	28.4	44.4		31.2	31.1		30.7	31.6	
Employed full time	18.4	18.1	20.5	20.5	13.9		20.8	15.5		16.3	20.2	
Unemployed	41.1	43.0	41.0	40.9	38.9		37.0	46.0		44.4	38.1	
Prefer not say	9.4	7.4	7.7	10.2	2.8		11.0	7.4				
**SNAP enrollment**						0.06			0.75			0.09
Never enrolled	22.7	26.9	18.0	21.6	16.7		22.5	23.0		28.8	17.3	
Past enrolled	13.1	9.4	20.5	18.2	8.3		13.9	12.2		11.1	14.9	
Enrolled (<1 year)	14.0	11.4	15.4	11.4	30.6		15.6	12.2		12.4	15.5	
Enrolled (≥1 year)	50.2	52.4	46.2	48.9	44.4		48.0	52.7		47.7	52.4	

Shown are the frequencies as percentages; due to missing data or specified “prefer to not answer”, some categories may not sum to 100% of the study sample. ^a^ Language of survey administration. ^b^ “Married” includes those living with a partner. “Single” includes those who are separated from a partner, divorced, or widowed. ^c^ “Employed part time” includes those who work “up to 20 h per week for pay”. “Employed full time includes those working “21–40 h per week for pay”. ^d^ Those included in the “Other” category self-identified as Asian, Native Hawaiian or Pacific Islander, American Indian or Alaska Native, or Multiple race/ethnicities. The remaining 2.8% of respondents (*n* = 9) preferred to not report race/ethnicity. ^e^ Those who responded agreed with one or both of the following items were categorized as “food insecure” (else were categorized as “food secure”): (1) Since the start of COVID (February 2020), we worried whether our food would run out before we got money to buy more, and/or (2) since the start of COVID (February 2020), the food we bought just did not last and we did not have money to get more. ^f^ *p* value determined by using χ^2^ test. Bold font indicates statistical significance.

**Table 2 nutrients-14-04947-t002:** MA WIC Caregiver Perceptions of FV Accessibility and Purchasing Post-CVB-Increase by Race/Ethnicity, Average Market Travel Time, and Food Security Status.

FV Accessibility and Purchasing Characteristics	All	By Race/Ethnicity					By Market Travel Time			By Food Security Status ^b^			
		Hispanic	NH Black	NH White	Other ^c^	*p* ^a^	<15 min	≥15 min	*p* ^a^	Food Secure	Food Insecure	*p* ^a^	
	(*n* = 321)	(*n* = 149)	(*n* = 39)	(*n* = 88)	(*n* = 36)		(*n* = 173)	(*n* = 148)		(*n* = 153)	(*n* = 168)		
**Market & FV access**													
**Where do you do most of your shopping?**						0.09			**0.002**			**0.01**	
Supercenter	25.2	28.2	28.2	20.5	22.2		17.3	34.5		24.2	26.2		
Supermarket	68.9	62.4	69.2	78.4	72.2		76.9	59.5		73.9	64.3		
Other	5.9	9.4	2.6	1.1	5.6		5.8	6.1		2.0	9.5		
**How do you usually get to the store?**						**0.005**			0.14			0.06	
Car	69.5	57.7	66.7	85.2	80.6		74.0	64.2		76.5	63.1		
Public transit	8.4	12.1	7.7	5.7	2.8		5.2	12.2		5.9	10.7		
Walk	5.6	7.4	10.3	1.1	2.8		4.6	6.8		3.9	7.1		
Catch a ride	15.6	22.2	15.4	8.0	11.1		15.6	15.5		13.7	17.3		
Other	0.9	0.7	.	.	2.8		0.6	1.4		.	1.8		
**Does your store sell the FV your family likes?**						0.39			0.37			0.88	
Yes	96.9	96.0	100.0	96.6	100.0		97.7	96.0		96.7	97.0		
No	3.1	4.0	.	3.4	.		2.3	4.1		3.3	3.0		
**Are the FV prices at your store reasonable?**						0.47			0.45			**<0.0001**	
Yes, reasonable	72.0	77.9	64.1	68.2	69.4		76.3	66.9		83.0	61.9		
No, expensive	28.0	22.2	35.9	31.8	30.6		23.7	33.1		17.0	38.1		
**FV purchasing**													
**In the past month, could you get enough FV for your family?**						0.21			0.06			**<0.0001**	
Yes	61.7	62.4	66.7	62.5	50.0		63.6	59.5		73.2	51.2		
No	38.3	37.6	33.3	37.5	50.0		36.4	40.5		26.8	48.8		
**In the past month, how much of your own money did you spend on household groceries?**						**0.009**			0.68			**0.0002**	
$0–25	13.4	10.1	10.3	22.7	5.6		14.5	12.2		20.9	6.6		
$26–50	24.9	24.2	20.5	28.4	27.8		24.3	25.7		28.8	21.4		
$51–100	15.6	18.8	20.5	14.8			17.9	12.8		10.5	20.2		
$101–200	31.5	28.9	35.9	27.3	44.4		29.5	33.8		28.1	34.5		
>$201	14.6	18.1	12.8	6.8	22.2		13.9	15.5		11.8	17.3		

Shown are the frequencies as percentages; due to missing data or specified “prefer to not answer”, some categories may not sum to 100% of the study sample. ^a^ *p* values determined by using χ^2^ test. Bold font indicates statistical significance. ^b^ Those who responded agreed with one or both of the following items were categorized as “food insecure” (else were categorized as “food secure”): (1) Since the start of COVID (February 2020), we worried whether our food would run out before we got money to buy more, and/or (2) since the start of COVID (February 2020), the food we bought just did not last and we did not have money to get more. ^c^ Those included in the “Other” category self-identified as Asian, Native Hawaiian or other Pacific Islander, American Indian or Alaska Native, or Multiple race/ethnicities. The remaining 2.8% (*n* = 9) preferred to not report race/ethnicity.

**Table 3 nutrients-14-04947-t003:** Frequency of Perceived Improved FV Behaviors and Satisfaction Outcomes, and Estimated Adjusted Probability of Reporting Improved Outcomes, by Race/Ethnicity, Average Market Travel Time, and Food Security Status.

		by Race/Ethnicity				by Market Travel Time		by Food Security Status ^c^	
How Did the CVB Increase Affect Perceived Changes in FV Behavior and Satisfaction?	All	NH Black vs. NH White		Hispanic vs. NH White		<15 min vs. ≥15 min		Food Insecure vs. Secure	
	**% ^a^**	**RR** **(95% CI) ^b^**	** *p* **	**RR** **(95% CI) ^b^**	** *p* **	**RR** **(95% CI) ^b^**	** *p* **	**RR** **(95% CI) ^b^**	** *p* **
**FV affordability**									
Decreased out-of-pocket FV cost	70.4	0.87 (0.69, 1.09)	0.22	**0.86** **(0.74, 1.00)**	**0.05**	1.01 (0.87, 1.16)	0.94	0.96 (0.83, 1.11)	0.57
**FV behaviors**									
Increased amount FV offered to children	70.1	0.80 (0.58, 1.09)	0.15	1.11 (0.94, 1.32)	0.22	1.03 (0.90, 1.19)	0.67	0.97 (0.84, 1.12)	0.70
Increased amount FV caregivers personally consume	63.2	0.88 (0.64, 1.21)	0.44	1.04 (0.85, 1.28)	0.68	0.96 (0.81, 1.14)	0.64	1.03 (0.87, 1.22)	0.73
Increased amount FV purchased	71.0	0.84 (0.63, 1.11)	0.22	1.06 (0.91, 1.24)	0.44	1.03 (0.89, 1.19)	0.67	0.92 (0.81, 1.06)	0.26
**Form FV purchased**									
Increased amount fresh FV	72.6	0.83 (0.63, 1.11)	0.21	1.01 (0.86, 1.18)	0.93	1.08 (0.94, 1.24)	0.26	1.05 (0.91, 1.20)	0.52
Decreased or same amount canned FV	68.9	1.13 (0.92, 1.39)	0.25	0.96 (0.80, 1.16)	0.70	1.12 (0.95, 1.32)	0.16	0.93 (0.79, 1.10)	0.39
Decreased or same amount frozen FV	65.7	1.10 (0.87, 1.40)	0.41	1.00 (0.82, 1.20)	0.96	1.08 (0.92, 1.27)	0.35	0.98 (0.92, 1.16)	0.79
**Satisfaction**									
Increased satisfaction with FV taste and quality ^d^	55.5	0.87 (0.58, 1.31)	0.51	1.07 (0.84, 1.38)	0.58	1.01 (0.82, 1.25)	0.89	0.98 (0.80, 1.21)	0.85
Increased satisfaction with WIC food package ^d^	37.1	0.90 (0.58, 1.39)	0.64	**0.66** **(0.47, 0.94)**	**0.02**	0.97 (0.72, 1.31)	0.85	0.97 (0.72, 1.31)	0.22

Note: Effective sample size is 304, as seventeen responses were excluded due to missing data ^a^ Shown are the frequencies as percentages. ^b^ Poisson linear regression models, which adjusted for caregiver age and educational attainment were used to estimate the relative risk (probability) of reporting perceived improvement for each FV behavior outcome by the independent variables of race/ethnicity (NH Black vs. NH White, Hispanic vs. NH White), average travel time to market (<15 min vs. ≥15 min), and food security status (insecure vs. secure). Bold font indicates statistical significance. ^c^ Those who responded agreed with one or both of the following items were categorized as “food insecure” (else were categorized as “food secure”): (1) Since the start of COVID (February 2020), we worried whether our food would run out before we got money to buy more, and/or (2) since the start of COVID (February 2020), the food we bought just did not last and we did not have money to get more. ^d^ Calculated as the difference between satisfaction (reported on 5-point scale) post- vs. pre-CVB-increase.

## Data Availability

Measures, data, and code are publicly available on Open Science Framework (https://osf.io/86c23/?view_only=242ebf5be24544da991077356bb29cd3, accessed on 21 November 2022).

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
