# Peer review of "Caregivers’ Perceived Impact of WIC’s Temporary Cash-Value Benefit (CVB) Increases on Fruit and Vegetable Purchasing, Consumption, and Access in Massachusetts"

_nutrients, 2022, doi:10.3390/nu14234947_

Round 1
Reviewer 1 Report
In the manuscript „Caregivers’ perceived impact of WIC’s temporary Cash-Value Benefit (CVB) increases on fruit and vegetable purchasing, consumption, and access in Massachusetts” the Authors tried to evaluate caregiver perceptions of the Special Supplemental Nutrition Program for Women, Infants, and Children in Massachusetts regarding fruit and vegetable accessibility post Cash Value Benefit. Moreover, the Authors tried to perceive the impact of increase of the Cash Value Benefit on food and vegetable behaviors and satisfaction with the food package of the program. In addition, the attempt to assess potential disparities in perceived impact by race and ethnicity, market access, and food security status is to be commended.
The manuscript provides valuable information. However, I have a few questions.
Abstract;
There are too many keywords. Look at the Instructions for Authors.
Introduction;
The Authors wrote in the Introduction that “Today, over six million women, infants, and children participate in WIC, from a pool of nearly 11 million eligible”. I am thinking why only six million since the program delivers such benefits? Do you know explanation?
Materials and methods;
The text message was sent on a number of cell-phone, or social media? What way did you use?
Author Response
- Comment 1: (Abstract) There are too many keywords. Look at the Instructions for Authors.
- Response: We thank the reviewer for their attention to detail. We can confirm we now have 10 key terms (within the specified limit of 3-10), which are listed below for reference.
- (1) Special Supplemental Nutrition Program for Women, Infants, and Children (WIC);
- (2) Cash-Value Benefit (CVB);
- (3) Fruits and Vegetable Consumption;
- (4) Fruit and Vegetable Affordability;
- (5) Nutrition Policy;
- (6) Early Childhood;
- (7) Access to Healthy Foods;
- (8) Healthy Nutrition;
- (9) Food Assistance;
- (10) Low-Income Population
- Response: We thank the reviewer for their attention to detail. We can confirm we now have 10 key terms (within the specified limit of 3-10), which are listed below for reference.
- Comment 2: (Introduction) The Authors wrote in the Introduction that “Today, over six million women, infants, and children participate in WIC, from a pool of nearly 11 million eligible”. I am thinking why only six million since the program delivers such benefits? Do you know explanation?
- Response: We thank the reviewers for underscoring this important question, which we agree is a critical point of discussion for the introduction. To address this, we have added the following text to the introduction, “Though nearly 11 million women, infants, and children are eligible for WIC, only about one-in-two of those eligible participate in the program, underscoring a critical gap in coverage[8]. Among known barriers to optimal retention and redemption (e.g., dissatisfaction or cultural mismatch with the food package[20,21], or lack of childcare, time, or transit to make WIC appointments[22-24]), low CVB value was commonly cited pre-pandemic as a reason for early drop off from WIC while still eligible[10].”
- Comment 3: (Materials and methods) The text message was sent on a number of cell-phone, or social media? What way did you use?
- Response: We thank the reviewer for seeking clarification. We only used text messages to recruit participants; we did not use social media. Our partners at the MA WIC State Office texted enrolled WIC participants an invite to join the study via linked survey. This method, as opposed to advertisements via social media, helped ensure we were recruiting and surveying verified WIC-enrolled caregivers.
Reviewer 2 Report
This is interesting work and good effort,one suggestion is your conclusion is very weak and need regions this is not a summary , add limit ions, future work….
Author Response
- Comment 1: This is interesting work and good effort,one suggestion is your conclusion is very weak and need regions this is not a summary , add limit ions, future work…
- Response: We thank the reviewer for their feedback. Aligning with the reviewer’s recommendation, we do critically interpret our findings in light of other relevant studies through the discussion section’s paragraphs four and five (lines 221-245), as well as offer recommendations for future research studies and reflect on both the strengths and limitations of the current study.
- Specifically, in terms of future work, we urge researchers to (1) more thoroughly evaluate the role that CVB increases may play in influencing WIC retention and redemption of benefits more directly, (2) leverage redemption data and/or dietary intake data to answer these questions more fully, and (3) reexamine the potential roles of race/ethnicity, market travel time, and food security status as moderators in perceived CVB benefit.
- We critically examine the strengths and limitations of our study through the following text in the discussion section: “Strengths of this study include the participation of a random sample of diverse MA-WIC-enrolled caregivers, as well as the fact that we were able to roll out the survey soon after the CVB increase had gone into effect, so that participants would have better recall about the change. As a preliminary study of this policy change, in which we quickly deployed a low-burden survey to capture participants’ perceptions, our study also has several limitations. First, because the survey assessed caregivers’ perceptions of the pro-gram, we cannot say for sure that the improvements seen in this study translate to actually higher FV benefit redemption and consumption. Future research should leverage redemption data and/or dietary intake data to answer these questions more fully. The cross-sectional design also relies on participant recollection (regarding WIC experiences day of survey vs. year prior), as data pre-CVB increase was not collected. Our response rate was also lower than anticipated, which may have been due to our use of text-based recruitment and online survey tools[34]; however, these methods were selected to help maintain participant privacy, reduce risk of social desirability bias, and increase reach (e.g., offering flexible response languages and times). In spite of these challenges, we were still able to recruit a representative sample, compared to MA and national WIC populations[35]. However, the small subsample sizes following stratification by race/ethnicity, market travel time, and food security status limited our power to detect differences by groups; these comparisons should be repeated in future studies with larger samples.”
- It is true that our conclusion is brief; we dedicate this short paragraph as a space to underscore key takeaways for readers. This matches prior Nutrients publications and aligns with Author Guidelines (which state the conclusion “is not mandatory but can be added to the manuscript if the discussion is unusually long or complex”).
- Response: We thank the reviewer for their feedback. Aligning with the reviewer’s recommendation, we do critically interpret our findings in light of other relevant studies through the discussion section’s paragraphs four and five (lines 221-245), as well as offer recommendations for future research studies and reflect on both the strengths and limitations of the current study.
Reviewer 3 Report
Some of the journal titles in your references are not formatted correctly.
Author Response
- Comment 1: Some of the journal titles in your references are not formatted correctly.
- Response: We appreciate the reviewer’s attention to detail. We have reviewed and revised the references.